# The magnitude of, and factors associated with, loss to follow-up among patients treated for sexually transmitted infections: a multilevel analysis

Mache Tsadik,[1] Yemane Berhane,[2] Alemayehu Worku,[3] Wondwossen Terefe[1]

## ABSTRACT

**Objectives** The loss to follow-up (LTFU) among patients attending care for sexually transmitted infections (STIs) in Sub-Saharan Africa is a major barrier to achieving the goals of the STI prevention and control programme. The objective of this study was to investigate individual- and facility-level factors associated with LTFU among patients treated for STIs in Ethiopia.

**Methods** A prospective cohort study was conducted among patients attending care for STIs in selected facilities from January to June 2015 in the Tigray region of Ethiopia. LTFU was ascertained if a patient did not present in person to the same facility within 7 days of the initial contact. Multilevel logistic regression was used to identify factors associated with LTFU.

**Results** Out of 1082 patients, 59.80% (647) were LTFU. The individual-level factors associated with LTFU included having multiple partners (adjusted OR (AOR) 2.89, 95% CI 1.74 to 4.80), being male (AOR 2.23, 95% CI 1.63 to 3.04), having poor knowledge about the means of STI transmission (AOR 2.08, 95% CI 1.53 to 2.82), having college level education (AOR 0.38, 95% CI 0.22 to 0.65), and low perceived stigma (AOR 0.60, 95% CI 0.43 to 0.82). High patient flow (AOR 3.06, 95% CI 1.30 to 7.18) and medium health index score (AOR 2.80, 95% CI 1.28 to 6.13) were facility-level factors associated with LTFU.

**Conclusions** Improving patient retention in STI follow-up care requires focused interventions targeting those who are more likely to be LTFU, particularly patients with multiple partners, male index cases and patients attending facilities with high patient flow.

[1]College of Health Science, Mekelle University, Tigray, Ethiopia
[2]Addis Continental Institute of Public Health, Addis Ababa, Ethiopia
[3]Addis Ababa University, Addis Ababa, Ethiopia

**Correspondence to**
Dr Mache Tsadik;
adhana2008@gmail.com

## Strengths and limitations of this study

► This study generated useful information that can help improve the clinical and public health interventions in the management of sexually transmitted infections (STIs) and in reducing loss to follow-up (LTFU).

► The study used a prospective cohort design that has the potential to minimise biases related to other observational study designs.

► An advanced statistical analytic model was used that allows simultaneous examination of individual and facility level factors that can affect the follow-up of patients attending STI care.

► The specific reasons for LTFU were not identified because there was no established participant tracing mechanism for patients who were LTFU in the context of this study.

► The study included only STI patients attending public health facilities; thus inference to patients seeking care in private facilities requires careful consideration of the local context.

symptoms and is often implemented at the primary healthcare level. As PN prevents risk of reinfection among regular partners and new infection among casual partners, treating partner(s) of the index case as early as possible is critical.[5] Thus it is important to follow index cases to assure compliance with treatment, and ascertain partner notification status[6 7] in order to effectively reduce the burden of STIs.[5 8] However, loss to follow-up (LTFU) has remained one of the challenges to effectively implementing the existing treatment and preventive strategies, and information on LTFU from Sub-Saharan Africa is scanty.

A number of factors have been linked to LTFU in the management of STI cases. LTFU is more likely among males,[9] single (not in union) index cases,[10] individuals

## INTRODUCTION

The prevalence and incidence of sexually transmitted infections (STIs) in Sub-Saharan Africa are among the highest in the world.[1] Due to shortcomings related to laboratory capacity, STI prevention and control programmes adapted a syndromic management in which partner notification (PN) is a key component of treatment package.[2–4] Syndromic management is a highly sensitive approach which responds to patients'

with a low level of education compared with a higher level of education,[11] individuals with poor knowledge of STIs,[12] and individuals who do not intend to notify partners.[13] Another factor is a reluctance to return to the same facility for follow-up due to fear of negative judgments.[14] On the provider side, poor quality of health services including inadequate patient education and lack of follow-up advice,[15] a judgmental approach of care providers, and a lack of privacy and confidentiality contribute to LTFU.[10 16–18]

This study was conducted in public health facilities of the Tigray region, North Ethiopia where little information is known about the magnitude of LTFU among patients treated for STIs and the associated factors. The public health facilities implement the national syndromic management protocol and treatment is provided at low cost to the individual. Therefore, the aim of this study was to investigate individual and facility-level factors associated with LTFU among STI patients attending public health facilities in North Ethiopia.

## METHODS
The study was conducted in public health facilities of the Tigray regional state, North Ethiopia. We selected health facilities with a monthly patient load (STIs) of five and above in order to make the research project manageable with the resources available for the study. Thus, of the 108 public health facilities in the study area, 27 fulfilled the study selection criteria. According to the national guideline, STI syndromes include vaginal discharge, urethral discharge, genital ulcer, lower abdominal pain, scrotal swelling, inguinal bubo and neonatal conjunctivitis.[19] Thus patients presenting with complaints such as burning sensation, genital discharge, genital ulcer and other related symptoms were treated as cases of STI using the syndromic management protocol that was adapted from the WHO generic protocol.

We conducted a prospective cohort study among patients attending public health facilities for STI care. Self-referred patients with one or more of the STI syndromes and who had sexual intercourse within 3 months preceding the study were recruited as study subjects. All patients that came seeking treatment for STI related symptoms during the study period were included in the study. The research team in collaboration with care providers ensured that patients received follow-up advice and appointment. Patients were verbally consented after they received routine care and informed of their right to decline any time. A baseline interview was then conducted to collect relevant information from each patient by trained research nurses in a private room. Patients were informed to notify their partners and return for follow-up within 7 days. Patients received instructions and an information card containing details of the facility room they should return to and contact details of the research assistant to facilitate their follow-up visit. A minimum of two research assistants was also assigned to each health facility to deal with those patients who returned for their follow-up within the specified period.

We calculated the sample size for LTFU using the assumptions of 50% LTFU by unmarried individuals, an OR of 1.5 at 95% CI and 80% power with a non-response rate of 10% which gave us a total sample size of 889. However, we had another objective which aimed to determine the predictors of PN among the same population, and obtained a sample size of 1095 though the eligible study participants enrolled in the study were 1082. In this regard, the following assumptions were considered: 40% of married individuals who notified partners[20] and an OR of 1.5, 80% power with 95% CI and 20% non-response rate. Thus, we took the pooled sample to increase power.

The study tool was developed by reviewing the relevant literature and then adapting it to the context of our study. At the individual level, the tool comprises sociodemographic, behavioural and psychosocial components. Some of the factors considered at the facility level were distance, trained providers, availability of treatment guidelines, patient flow, and health index score. The tool was pre-tested before the actual use in facilities not included in the study. The study measurements were defined and described in table 1. According to the Health Management Information System (HMIS) report of Tigray Regional Health Bureau, public health facilities were classified into three levels using the health facility index score as low (<50), medium (50–74.9) and high (≥75).[21] However, among the facilities selected for this study no health facility belonged in the 'low' category.

### Statistical analysis
LTFU was ascertained if the index case failed to return to the same health facility within 7 days of the initial clinic visit; patients who were not LTFU were referred to as 'in follow-up care'. LTFU was categorised as 'Yes' for those LTFU and 'No' for those retained in care.

Both individual and facility-level variables were described and presented using a simple frequency table. Before multivariate analysis was performed, Pearson's $\chi^2$ square tests were used to check for the crude association between the dependent variable (LTFU) and individual and facility level variables. Then, all independent variables with p values smaller than the significance level (0.05) were entered into the model.

In multilevel analyses, a null model with no covariates was used to assess the presence of significant clustering in LTFU. For individual-level factors the analysis considered sex, marital status, education, the number of partners, type of partnership, knowledge of STI transmission methods and complications, and perceived stigma. Facility-level variables included health facility index and STI patient flow. The command 'xtmelogit' was used to fit a mixed-effect multilevel logistic regression model and the relationship between the dependent variable and each of the independent variables (ie, fixed effects) were assessed using odds ratios and their confidence intervals. To evaluate the significance of facility-level clustering of the

**Table 1** Description and measurements of variables in the models, North Ethiopia, 2015

| Variable | Description | Measurement |
|---|---|---|
| Type of partnership | The relationship of index case with sexual partner (regular—if there is an ongoing relationship for >3 months; casual—if the relationship is <3 months). | Regular, casual |
| STI transmission | Patient's knowledge of STI transmission: mean (SD) score 2.74 (1.69), five items:<br>1. Unprotected sexual intercourse<br>2. Mother to child during birth<br>3. Injury by sharp materials (needle, blade)<br>4. Blood transfusion<br>5. Breast feeding<br>*Response category (yes, no, I don't know) (Cronbach's α=0.77) | ≥Mean=good;<br><mean=poor |
| STI symptoms | Patient's knowledge of STI symptoms: mean (SD) score 4.03 (2.04), six items:<br>1. Vaginal discharge<br>2. Itchiness in genitalia<br>3. Pain/swelling in the groin<br>4. Pain on passing urine<br>5. Ulcers in the genitalia<br>6. Eye discharge in newborn<br>*Response category (yes, no, I don't know) (Cronbach's α=0.84) | ≥Mean=good;<br><mean=poor |
| STI prevention | Patient's knowledge of STI prevention: mean (SD) score 4.62 (1.00), five items:<br>1. Abstinence<br>2. Having single partner<br>3. Avoid sex with risk partners<br>4. Use of condom<br>5. Early treatment<br>*Response category (yes, no, I don't know) (Cronbach's α=0.81) | ≥Mean=good;<br><mean=poor |
| STI complication | Patient's knowledge of STI complication: mean (SD) score 2.01 (1.84), five items:<br>1. Cancer of cervix<br>2. Stillbirth<br>3. Abortion<br>4. Infertility<br>5. Ectopic pregnancy<br>*Response category (yes, no, I don't know) (Cronbach's α=0.82) | ≥Mean=good;<br><mean=poor |
| Perceived stigma | Patient's perceived stigma to PN: mean (SD) score 12.92 (2.52), four items:<br>1. Referring a partner for STI diagnosis and treatment is shameful<br>2. Attending a health facility for STI treatment is embarrassing<br>3. A good man/woman goes to a health facility for STI treatment<br>4. A good man/woman notifies his/her partner<br>*Response category (very likely, likely, unlikely, very unlikely)<br>(Cronbach's α=0.73) | ≥Mean=high;<br><mean=low |
| HFI* | Health facility index:<br>It is a ranking and prioritisation of health services. It is also an aggregate score of health facility performance such as availability of adequate resources, implementation plan, client satisfaction, community service, etc | High (>75), medium (50–74.9), low (<50) |
| HFS* | Health facility setting | Urban, rural |
| Distance‡ | Walking distance of health facility from home | ≤1 hour's walk, >1 hour's walk |
| STI trained† | Availability of trained care provider in STIs | Yes, no |
| STI patient flow* | Annual STI patient flow to health facility: mean (SD) score 166.26 (85) | ≥Mean=high;<br><mean=low |
| Guideline† | Availability of guidelines | Yes, no |

*Health management information system (HMIS) = data from regional health bureau.
†Facility assessment.
‡Patient's interview.
According to HMIS report, none of the selected health facilities were in the 'low' category.
PN, partner notification; STI, sexually transmitted infection.

dependent variable (ie, random effects), log-likelihood ratio tests were employed. Collinearity between variables was assessed by looking at the values of variance inflation factors (VIF). VIF >10 is assumed to be suggestive of the presence of multicollinearity. However, in this study the mean correlation value in the fitted model was 2.02.

## RESULTS

A total of 1082 patients who received STI care in selected health facilities were enrolled in the study; of these patients, 647 (59.80%, 95% CI 56.88% to 62.72%) who did not return for follow-up care within 7 days were categorised as LTFU.

### Patient and facility-level characteristics

Patient and facility level characteristics of the study sample are presented in table 2. The mean (SD) age of the cohort population was 26.4 (7.6) years. More than 50% of the patients who presented with STIs had at least a high school level of education. A substantial number of patients (41.22%) reported casual partnerships. The majority of the health facilities (73.38%) were located in urban settings. About three-fourths of patients reported residing within 1 hour walking distance from the health facility they visited. A large proportion of LTFU (73.2%) was observed in health facilities with high patient flow.

### Multilevel logistic regression analysis

After controlling the potential confounders at the individual and facility level, the odds of LTFU were greater among index cases with multiple partners (adjusted OR (AOR) 2.89, 95% CI 1.74 to 4.80), males (AOR 2.23, 95% CI 1.63 to 3.04), individuals with poor knowledge of STI transmission (AOR 2.08, 95% CI 1.53 to 2.82), and individuals with poor knowledge of STI complications (AOR 1.56, 95% CI 1.15 to 2.12). LTFU was less likely among better educated individuals (AOR 0.38, 95% CI 0.22 to 0.65) and those with perceived low stigma (AOR 0.60, 95% CI 0.43 to 0.82). LTFU was more likely among patients who received care in facilities with high patient flow (AOR 3.06, 95% CI 1.30 to 7.18) and among patients who received care in facilities with a medium health index score compared with the highest index score (AOR 2.80, 95% CI 1.28 to 6.13) (table 3).

## DISCUSSION

About two-thirds (59.8%) of STI patients were LTFU in this study. The individual level factors associated with an increased likelihood of LTFU were being male, having multiple sexual partnerships, and having poor knowledge about the means of STI transmission and their complications. Those individuals who achieved a higher level of education and reported low perceived stigma were less likely to be LTFU. The odds of LTFU were greater among patients seen in health facilities with a medium health index score and in facilities with high patient flow.

The level of LTFU among STI patients in our study is similar to the findings of other studies.[10 18] Early response to treatment within a period of a week and fear of stigma on return to follow-up were identified as factors contributing to LTFU.[22 23] Our study clearly indicates the potential for reinfection is quite high and that may in turn facilitate the development of drug resistant STIs. In addition, since a substantial proportion of cases reported multiple sexual partnerships, those untraceable and probably re-infected would continue to spread STIs in the community. This high proportion of LTFU is a major challenge to STI prevention and control efforts and needs to be addressed urgently.

The study identified a number of independent LTFU predictors both in individual and facility-level factors. In this study, males were more likely to be LTFU compared with females. This finding was consistent with studies conducted in Uganda,[24] Malawi[25] and South Africa.[26] As suggested by Geng et al, males are more likely to use substances that potentially decreases their adherence to follow-up care.[27] Males also report high risky sexual behaviours that may potentially attribute to LTFU because of linked stigma.[28]

The decreased likelihood of LTFU among educated individuals in this study is consistent with a previous study.[10 29 30] This may suggest that education is an important factor in adhering to medical care.[31] Highly educated individuals have a greater knowledge of STIs; such individuals are therefore less likely to be LTFU[12] possibly because of fear of subsequent complications. The motivation to attend follow-up care among educated individuals may also be associated with a greater understanding of the potential benefits.

LTFU was less likely among patients with low perceived stigma compared with those with high perceived stigma. This may show that individuals with low perceived stigma are confident enough to notify partners and have the courage to return for follow-up.[32] The stigma linked to STIs reduces the motive and willingness of index cases to notify partners and results in greater LTFU.[33] Fear of the healthcare provider's judgmental reactions during follow-up care and embracement negatively affects follow-up considerably in low income settings.[34]

LTFU was higher among patients from facilities with a high patient flow in the present study. A similar observation was reported previously in the HIV treatment setting where high patient load was associated with a high proportion of LTFU.[35] The high patient load might limit the provider's time to provide adequate care and that could potentially influence the index cases to LTFU.[36] The odds of LTFU were also lower among patients attending facilities with a high index score in our study. Similarly, a patient focused study conducted in Nigeria has shown that patients who received high quality care were less likely to be LTFU.[37] This may indicate that high quality care motivates patients to remain in follow-up care.

This study has some important limitations. First, the reasons for LTFU were not documented as this

**Table 2** Profile of study subjects, North Ethiopia, 2015

| Characteristic | Loss to follow-up | | Pearson's $\chi^2$ (p value) |
|---|---|---|---|
| | **Yes (%)** | **No (%)** | |
| Gender | | | 0.001 |
| Female | 304 (50.7) | 295 (49.3) | |
| Male | 343 (71.0) | 140 (29.0) | |
| Age | | | 0.930 |
| <25 years | 326 (59.9) | 218 (40.1) | |
| ≥25 years | 321 (59.6) | 217 (40.4) | |
| Education | | | 0.001 |
| Illiterate | 140 (73.6) | 50 (26.4) | |
| Primary | 197 (60.8) | 127 (39.2) | |
| Secondary | 219 (55.1) | 178 (44.9) | |
| College | 91 (53.2) | 80 (46.8) | |
| Marital status | | | 0.152 |
| Married | 264 (51.2) | 252 (48.8) | |
| Single | 383 (67.7) | 183 (32.3) | |
| Residence | | | 0.510 |
| Urban | 486 (60.4) | 319 (39.6) | |
| Rural | 161 (58.1) | 116 (41.9) | |
| Type of partnership | | | 0.001 |
| Regular | 346 (54.4) | 290 (45.6) | |
| Casual | 301 (67.5) | 145 (32.5) | |
| No. partners in last 3 months | | | 0.001 |
| One | 533 (56.8) | 405 (43.2) | |
| Two or more | 114 (79.2) | 30 (20.8) | |
| Perceived stigma to PN | | | 0.003 |
| High | 248 (63.6) | 142 (39.4) | |
| Low | 399 (57.6) | 293 (42.4) | |
| Types of STI syndromes | | | 0.069 |
| Vaginal discharge | 233 (44.21) | 294 (55.79) | |
| Urethral discharge | 128 (34.69) | 241 (65.31) | |
| Genital ulcer | 32 (40.51) | 47 (59.49) | |
| Lower abdominal pain | 28 (41.79) | 39 (58.21) | |
| Others | 14 (35.00) | 26 (65) | |
| Distance from health facility | | | 0.443 |
| <1 hour walk | 450 (54.6) | 374 (45.4) | |
| ≥1 hour walk | 197 (76.4) | 61 (23.6) | |
| Health facility index score | | | 0.001 |
| High | 176 (50.8) | 170 (49.2) | |
| Medium | 471 (64.0) | 265 (36.0) | |
| Health facility setting | | | 0.554 |
| Urban | 479 (60.3) | 315 (39.7) | |
| Rural | 168 (58.3) | 120 (41.7) | |
| Patient flow to health facility | | | 0.001 |
| Low | 324 (48.5) | 344 (51.5) | |
| High | 111 (26.8) | 303 (73.2) | |

p<0.05 was considered statistically significant.
PN, partner notification; STI, sexually transmitted infection.

**Table 3** Multivariable multilevel logistic regression analysis of individual and health facility level factors associated with loss to follow-up, North Ethiopia, 2015

| Characteristics | Category | AOR (95% CI) | p Value |
|---|---|---|---|
| Individual level variables | | | |
| Sex of index case | | | 0.001 |
| | Female | 1 (ref) | |
| | Male | 2.23 (1.63 to 3.04) | |
| Educational status | | | 0.001 |
| | Illiterate | 1 (ref) | |
| | Primary | 1.11 (0.71 to 1.74) | 0.633 |
| | Secondary | 0.89 (0.57 to 1.38) | 0.613 |
| | College | 0.38 (0.22 to 0.65) | 0.001 |
| Number of partners in last 3 months | | | 0.001 |
| | One | 1 (ref) | |
| | Two or more | 2.89 (1.74 to 4.80) | |
| Knowledge of STI transmission | | | 0.001 |
| | Good | 1 (ref) | |
| | Poor | 2.08 (1.53 to 2.82) | |
| Knowledge of STI complication | | | 0.004 |
| | Good | 1 (ref) | |
| | Poor | 1.56 (1.15 to 2.12) | |
| Perceived stigma | | | 0.002 |
| | High | 1 (ref) | |
| | Low | 0.60 (0.43 to 0.82) | |
| Health facility level variables | | | |
| Health facility index | | | 0.010 |
| | High | 1 (ref) | |
| | Medium | 2.80 (1.28 to 6.13) | |
| STI patient flow | | | 0.010 |
| | Low | 1 (ref) | |
| | High | 3.06 (1.30 to 7.18) | |
| Variance | | 0.74 | |
| ICC (%) | | 18.56 | |
| PCV (%) | | 27.88 | |
| Model fitness | | | |
| Log likelihood | | −577.92 | |
| AIC | | 1155.84 | |

AIC, Akaike information criterion; AOR, adjusted OR; ICC, intra class correlation coefficient; PCV, proportional change in variance; ref, reference; STI, sexually transmitted infection.

study did not have the resources to establish participant tracing mechanisms. Thus, patients who decided to do their follow-up care at another health facility because of STI linked stigma may have been considered as LTFU. Second, study participants who returned for follow-up after the scheduled time were not recorded. Third, since the study was conducted among self-referred patients from public health facilities, extrapolation of results to all STI patients should be made with caution since the factors related to LTFU among those seeking care in private facilities may not be similar. Behaviour related information was all self-reported and may have some reliability issues because of social desirability and recall bias.

Despite the stated limitations, this study is one of the few studies ever conducted on LTFU among patients treated for STIs in our setting. We believe the information reported will help to improve interventions being implemented to prevent and control STIs in contexts similar to our study. This study also employed an advanced method

of analysis that allowed the simultaneous analysis of individual and facility-level factors.

## CONCLUSION

Overall, the magnitude of LTFU among patients being treated for STIs is very high in North Ethiopia. The need for standardised follow-up care and cost effective tracing mechanisms is important to retain and trace STI patients who are at risk of LTFU.

**Acknowledgements** We would like to acknowledge Mekelle University and Addis Continental Institute of Public Health for the overall technical and material support. This research was partially funded by African Doctoral Dissertation Research Fellowship award (ADDRF) (Grant Number: ADF 015). We would also like to thank the Tigray Health Bureau and directors of the study health facilities for facilitating the conduct of this study. We are also grateful to all study participants, data collectors and supervisors without whom this work would not have been possible.

**Contributors** The study was designed by MT, YB, AW and WT. MT was responsible for data collection, analysis and drafting the manuscript. YB revised the study design and the manuscript. AW supervised the data collection and analysis, revised the manuscript, and contributed to interpretation of the analysis. WT participated in the analysis and interpretation of the data, as well as revised the manuscript. All authors have read and approved the final manuscript.

**Funding** The study is funded by Mekelle University and African Population Health Research Center (APHRC). The funders had no role in the gathering or analysis of the data and no role in the writing of the manuscript or the decision to submit for publication.

**Competing interests** None declared.

**Ethics approval** Health research Ethical review committee, Mekelle University.

**Provenance and peer review** Not commissioned; externally peer reviewed.

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
