## [Reviewer comments · BMJ Open]

ARTICLE DETAILS

TITLE (PROVISIONAL)	The magnitude and factors associated with loss to follow-up among patients treated for sexually transmitted infections: A multilevel analysis
AUTHORS	Tsadik, Mache; Berhane, Yemane; Worku, Alemayehu; Terefe, Wondwossen

VERSION 1 - REVIEW

REVIEWER	Sally Guttmacher New York University, NYC, USA
REVIEW RETURNED	30-Apr-2017

GENERAL COMMENTS	I think that this is an excellent paper and should be published as it is without revision.
--

REVIEWER	Martin Brinkhof Swiss Paraplegic Research, Nottwil, Switzerland
REVIEW RETURNED	04-May-2017

GENERAL COMMENTS	Loss to follow-up (LTFU) represents a major threat to the efficacy of health policy in the context of sexually transmitted infections (STIs), particularly in sub-Saharan Africa, where rates of LTFU are often excessive. LTFU commonly results in inefficient long-term treatment of STIs in both the index case as well the respective sexual partners, who are frequently not notified of the infection. As untreated individuals may engage in unprotected sex with future sexual partners, LTFU has been identified as a major driving force for the spread of STIs in African and other populations. This study from Ethiopia is liable to make an important contribution to the scant evidence-base regarding determinants of LTFU in context STIs. Such evidence is needed as to better identify targets for interventions that aim to effectively reduce LTFU, and following, improve treatment and prevention of STIs. The study provides good sample sizes to describe LTFU and I acknowledge the authors for investigating both highly relevant person characteristics as well as facility-level factors as potential determinants of LTFU. While providing as such good quality results, the paper unfortunately holds major shortcomings in structure, logic and writing, particularly impacting the Introduction and Discussion. These issues appear, in my view, resolvable in a revision. Specific comments 1) INTRODUCTION
---

The introduction needs some restructuring in order to strengthen the present focus on LTFU in STIs, mentioning partner notification (PN), the apparent motivation behind the principal study, only to contextualize and emphasize the relevance of the present research. Three successive paragraphs could concentrate on the following:

1.1) Introducing that LTFU presents a major problem to public health policies targeting the treatment and prevention of STIs using argumentation along the lines indicated above (thus containing topics of §2 and §3 from current manuscript).

1.2) Outline the established and potential risk factors for LTFU, with respect to durable STI treatment, grouped by individual (current lines 35-42) and facility-level factors (e.g., current lines 30-33). This section should be expanded on, with additional literature references as to provide a better rationale for the set of potential determinants of LTFU chosen in this present study (i.e., variables shown in Tables 1 and 3).

1.3) Brief outline of the setting of the current study, followed by aims and objectives. Details to the STI treatment guidelines and use of a syndromic management approach are better dealt with in the Methods section; the same applies to the use of multilevel modeling (lines 50-53).

2) METHODS

The Methods section also needs restructuring and some specific additional information to the study. Broader sections may successively include the following:

2.1) General description of setting. For instance, lines 37-41 should be moved to this section, also study site selection should be explained. It only became clear that the study involved 27 facilities only in Figure 1.

2.2) Description of presentation and identification of cases, including a brief explanation of operational diagnostics and treatment guidelines (syndromic management approach). Importantly, some detail needs to be provided regarding the different STIs that are being managed.

2.3) Details regarding study design and the recruitment and follow-up of study participants. Here procedures regarding informed consent should also be described in more detail. (“... volunteer to participate in the study ... “ in line 19 is too vague). The sample size calculation also needs clarification: it is unclear what is meant with “... an odds ratio of 1.5 at 95% CI ...” or “... a power of 80% with a response rate of 20% ...” (lines 43-46).

2.4) Details to the questionnaire tool that is being used are needed. In line with a renewed Introduction, a clear distinction between individual- and facility-level predictors for LTFU in STI treatment is desirable. Particularly the parameters in Table 1 need more explanation. To understand and judge the usefulness of the tools used, a brief summary is needed regarding the exact construct of each of the multi-item elements (e.g., STI transmission apparently containing 5 items; STI symptoms 6 items; etc) as well as available evidence from metric studies for their validity (in case as an

Appendix). Importantly, the index score for health facilities from the Regional Health Bureau needs explanation and specification, in particular attest confirm that LTFU was not included in the quality index.

2.5) Statistical analysis. All text regarding the definition of LTFU is best shifted to the start of this section (thus also lines 30-35). Then provide detail regarding the descriptive analysis, followed by the multivariable analysis. The hierarchical levels used in the current analysis, presumably facility- and patient-level, need better explaining. Furthermore, a rationale for the choice of predictor variables is needed, particularly why certain variables (e.g., Distance or Guideline; Table 1) were apparently excluded as potential predictors for LTFU in the data analysis (compared with Table 3). Also an explanation and justification for the use of binary-transformed scores (Table 1) in data analysis is lacking.

3) RESULTS

Main issues concern the following:

3.1) Figure 1 can be dropped. All relevant information is already available in Table 2, with the exception of the number of clinic sites where data were collected. See above, Methods issue I.

3.2) I strongly suggest to add 'type of STI' as a variable in descriptive Table 2, and additionally to the multivariable analyses in Table 3. This background info is highly relevant to the paper. It further seems highly plausible that risk for LTFU is related type of STI, also that STI type may thus represent a confounder for the some of the documented associations (Table 3), for instance for perceived stigma (not excluding other variables).

3.3) What is the reasoning behind using 'marital status' and 'No. of partners' as independent variables in analyses (Tables 2 & 3)? Aren't these variables highly correlated? This could use some justification in the Methods section, data analysis plan.

3.4) Table 2: 'No (%)' column for LTFU can be dropped; the header "Lost to follow-up" can also be dropped, presuming that the column "Entire cohort (%)" also refers to n (%) LTFU. It would be informative to add a column presenting a univariable comparison of LTFU rates (p-values from chi-square tests). There also needs to be a space in between n and (%).

3.5) Table 3: I strongly suggest to drop Models 1-3 from the table. The fact that effect sizes for individual and facility-level factors are only slightly mutually affected can be briefly mentioned in the Results. Test statistics should be given in detail, instead of using star '*' symbols, and row-wise aligned to the overall variable name (representing a global test, see next point; e.g. global test is for sex of index case, not male).

3.6) Table 3: Regarding significance testing, it is important to perform global tests to evaluate the contribution of a specific variable. This particularly concerns variables with more than one level, such as 'educational status': the present test statistic is only comparing 'Illiterate' vs. 'College+'. Instead a global test with 3 degrees of freedom is initially needed, followed by a post-hoc test in case of global significance. The post-hoc test needs to account for

multiple testing (e.g., using Bonferroni correction). In Stata, one can use commands “contrast” or “pwcompare”.

3.7) Table 3: Several potential predictor variables for LTFU do not appear in Table 3, e.g., ‘age’ and ‘distance to clinic’ (not excluding other omitted variables). Why is this the case? If a stepwise-backward selection procedure was used to reduce the initial full model to a final model, containing only significant (alpha 0.05) variables, then this needs to be explained in the Methods section. Also, for transparency it would be important to report the significance level at which these variables were dropped in the Results section.

4) DISCUSSION

4.1) The Discussion would benefit from a restructuring. Following an overall summary of the revised results, a contextual discussion of the overall rates of LTFU in persons with STIs are needed. In referencing other studies, it is highly relevant to specify the type of STI involved (e.g., HIV, gonorrhea, etc), as the dynamics of LTFU may vary. Next, a more focused discussion should follow on individual level and facility level variables, highlighting the Results of the multivariable analysis in Table 3 (taking point 4.2 into account). The limitations section also needs the author’s attention, as it contains inaccurate phrasings (e.g., line 23) and an implausibility e.g., line 24-26: the argument regarding “... lack of uniform definition on LTFU ... “ that seems to conflict with the uniform definition of the 1 week delay explained in the Methods section.

4.2) A fundamental problem in the current discussion is the use of perceived stigma (or further specific variables), in explaining the association of other variables, e.g. sex (lines 27-32) or education (lines 35-44) with LTFU. However, because inferences regarding associations were in the presence of perceived stigma, this argumentation is illogical. For example, in the multivariable analysis, the potential variation in LTFU in regards to sex or education, related to perceived stigma, has effectively been taken into account, as ‘stigma’ is one of the variables included in the model. Thus, the residual association must be explained by other factors. Summarizing, this will thus require a fundamental revision of substantial parts of the Discussion.

5) REFERENCES

The reference list needs careful checking for correctness of content (e.g., ref. 19 needs publisher information; ref. 15 contains two years of publication; ref. 29, name is Brinkhof, not Brinkhofs) and style (e.g., inconsistent use of capitals and semicolons throughout list; missing spaces, e.g., in ref 29 should read “Early loss ...” and “lower-income ...”; etc.)

6) FINAL REMARKS

The paper may benefit upon resubmission from a careful check on language and style by a native English speaker.

I hope that my comments are fair and useful in revising the manuscript.

VERSION 1 – AUTHOR RESPONSE

Comments from reviewer 1	Response to comment
Didn't leave comment	
Comments from reviewer 2	
1. Introduction	
1.1 Introducing problems related to lost to follow-up	Accepted and comments incorporated in the first paragraph
1.2 Outline risk factors to LTFU	Accepted and comments incorporated in the second paragraph
1.3 Setting of current study, followed by aims and objectives - Move syndromic management and multilevel to method part	Comment accepted and incorporated in the document
2) Methods	
2.1 • description of study setting and study site selection	Comment accepted • Study setting moved from line 37-41 into setting • Site selection explained in the first paragraph of the method part
2.2 Description and identification of case	Comment well taken and incorporated in the document
2.3 • Detail of study design, recruitment and follow-up • Procedure of informed concept • What is meant with odds ratio of 1.5 at 95% CI ...” or “... a power of 80% with a response rate of 20% ...” (lines 43-46).	Comments appreciated and addressed in the document • Sample size calculation and the assumptions taken are clarified in the document
2.4 • Details to the questionnaire particularly to the multi-item elements like Knowledge of STIs transmission..... • Explanation and specification of Index score to confirm LTFU was not included in the quality index	• The item questions detailed and incorporated in table 1 • As observed in the document, there are list of many components used to compute Index score such as number of staffs, availability of basic medical equipment, community based services (supporting HEW), patient satisfaction, number of patients served , readiness, cost of services etc. The LTFU case was not included as part of the quality index
2.5 Statistical analysis • Move definition of LTFU to the	• Done

first section  • Provide descriptive analysis followed by multivariate • Rationale for choice of predictor variables • Justification for use of binary transformation score 	 • Descriptive analysis is now brief and detail description is given about multilevel • We used Pearson chi-square value to screen variables to be fit in to subsequent models. As shown in table 2, those variables with p value (global test) less than 0.05 were considered for further analysis • For simplicity of analysis and interpretation
3 Results	
3.1 figure one can be dropped	 • Accepted
3.2 Strongly suggest to add “type of STIs”	 • Comment accepted and variable added into table 2, but not moved to further analysis since the Pearson's chi-square (p value) was greater than 0.05.
3.3 Justification for using marital status and number of partners , correlation	 • Comment appreciated, we checked for presence of correlation between independent variables using VIF. No variable was higher than VIF of 5 and VIF in the final model was 2.02
3.4 Add p value from chi-square test in table 2	Comment accepted and p value is added in the table 2
3.5 Strongly suggest to drop model 1-3 from table	Comment accepted and incorporated (see table 3)
3.6 Perform global test for specific variables with more than one level	Comment accepted and global test is performed We computed the p-value specifically for education which has multiple responses.
3.7 explain why several predictor variables don't appear in Table 3	Comment appreciated but the justification is the variables didn't appear in table 3 because they did not fulfill the criteria p value less than 0.05 to move to the next analysis .
4 Discussion	
4.1  • Summary of results • Contextual discussion of the overall rates of LTFU • More focused discussion on individual and multivariate • In accurate phrasing and implausibility in limitation section 	 • Summary done • Discussion on the overall rate of LTFU incorporated in the document • Discussion detailed than the previous version • Modified
4.2 Fundamental problem in use of perceived risk to explain the association of other variables	 • Comment accepted and discussion revised

5 References	 Accepted and corrections made
--------------	---

VERSION 2 – REVIEW

REVIEWER	Martin Brinkhof Swiss Paraplegic Research, Nottwil, Switzerland
REVIEW RETURNED	07-Jun-2017

GENERAL COMMENTS	The manuscript received major revision and the authors thus adequately addressed most of the suggestions and recommendations for revision in response to my first review. Remaining issues include the following. Major issues. 1) Introduction, §1: That loss to follow-up (LTFU) represents a major threat to the efficacy of health policy in the context of sexually transmitted infections (STIs) in sub-Saharan Africa could be expressed more clearly along the lines suggested in my previous review. The role of partner notification in preventing reinfection, also new-infections in context extra-marital or extra-pair relationships, has not been adequately worked out in context community-spreading of STIs in the face of LTFU. Terminologies “syndromic management” and “control programs” are not self-explaining and require better explanation in this context. 2) Methods: The issue of patient consent to study participation has once more not been addressed (see first review, specific comment 2.3) Further issues. 3) Suggest using terminology “loss to follow-up” instead of “lost to follow-up” to indicate the study topic. The use of abbreviation also needs harmonizing throughout paper 4) Introduction: Suggest to identify abbreviation PN for partner notification on first occasion. 5) Methods, lines 44-49: The sample size calculation is still not clear, in particular why an effect size related to PN instead of LTFU was used (also see first review, specific comment 2.3). As this study is observational and essentially involving convenience sampling, the additional data collection was not burdensome. The sample size calculation thus appears to be a minor issue. 6) Methods, page 6, lines 6-12: “Health Facility Index score,” should be more clearly defined, particularly because the reference (21) reflects a local report and is difficult to access. 7) Methods, page 6, lines 48-51, A short explanation of VIF is needed as well as an explanation and interpretation of the given model fit (2.02). 8) “Relief of Symptoms” needs to be expanded upon in the discussion section. Explanation would ideally include the type of STIs involved as well as the effectiveness of treatment and the corresponding time course of relief and cure.
--

	9) Table 2: “Types of STI syndromes” was close to significance and could have been included in the multivariable regression analysis, or at least as part of a sensitivity analysis. In any case, some further discussion is warranted. 10) Discussion, page 9, lines 20-23, It is unclear what is meant by “patients with high quality score.” Is this referring to the health facility or to individual patients characteristics? 11) Discussion, page 8, lines 39-41: The term stigma is wrongly used in this context. Probably "less concern" or similar is meant.
--	---

VERSION 2 – AUTHOR RESPONSE

Major issues	Response to comment
Comments from reviewer 2	
1. Introduction	
Paragraph1	
Terminology to Syndromic management	Accepted and comments incorporated in the first paragraph
The role of PN in preventing reinfection and infection in relation to LTFU	Comment incorporated in the revised version
2) Methods	
2.3 The issue of patient informed consent	Comments appreciated and addressed in the document
3)	
using terminology loss to follow-up instead of lost to follow up	Comment accepted and incorporated
Harmonized use of abbreviation	Comment accepted and corrected
4)	
Identify abbreviation PN on first occasion	Accepted and corrections made
5) Methods	
Why an effect size related to PN is	We appreciate the concern. We had two research

used instead of LTFU	objectives in the cohort population 1) magnitude and associated factors to LTFU 2) factors associated with partner notification Then, we calculated sample size for each. The sample size for LTFU was 895 whereas the sample size for partner notification was 1095. Since we used same population we took the pooled sample size to increase power.
6) Methods	
Health facility index score should be clearly defined	Explanation added in the revised version
7 Methods	
Short explanation of VIF is needed	Explanation added in the revised version
8. Relief of symptoms need to be expanded in discussion	The comment is accepted. Early response to treatment can better explain the “relief of symptoms”, because one of the indicators of early symptom is relief of symptom (the symptoms of the specific symptom will disappear within a week time starting from 3-4 days following treatment). So that we deleted “relief of symptoms”
9) Table 2	
Types of STI syndromes could have been included in multivariate analysis, need some further discussion	We appreciate the reviewers comment that p-value of 0.069 is marginally significant but it didn't fulfill the predetermined criteria to be included in multivariate logistic regression.
10) Discussion (page 9, line 20-23)	
Wha is meant by “patients with high quality score” ? Is this referring to HF or individual pts characteristics	The comment is accepted and corrected
11) Discussion	
The term stigma is wrongly used in page 8, line 39-41	Comment accepted